# Therapeutic Potential of *Hibiscus sabdariffa* Linn. in Attenuating Cardiovascular Risk Factors

**DOI:** 10.3390/ph16060807

**Published:** 2023-05-29

**Authors:** Syaifuzah Sapian, Asma Ali Ibrahim Mze, Fatin Farhana Jubaidi, Nor Anizah Mohd Nor, Izatus Shima Taib, Zariyantey Abd Hamid, Satirah Zainalabidin, Nur Najmi Mohamad Anuar, Haliza Katas, Jalifah Latip, Juriyati Jalil, Nur Faizah Abu Bakar, Siti Balkis Budin

**Affiliations:** 1Centre for Diagnostic, Therapeutic and Investigative Studies, Faculty of Health Sciences, Universiti Kebangsaan Malaysia, Kuala Lumpur 50300, Malaysia; syaifuzahsapian17@gmail.com (S.S.); aliasma515@gmail.com (A.A.I.M.); fatinfarhanajubaidi@gmail.com (F.F.J.); ejamdnor@gmail.com (N.A.M.N.); izatusshima@ukm.edu.my (I.S.T.); zyantey@ukm.edu.my (Z.A.H.); nurfaizah@ukm.edu.my (N.F.A.B.); 2Center for Toxicology and Health Risk Studies, Faculty of Health Sciences, Universiti Kebangsaan Malaysia, Kuala Lumpur 50300, Malaysia; satirah@ukm.edu.my (S.Z.); nurnajmi@ukm.edu.my (N.N.M.A.); 3Centre for Drug Delivery Technology, Faculty of Pharmacy, Universiti Kebangsaan Malaysia, Jalan Raja Muda Abdul Aziz, Kuala Lumpur 50300, Malaysia; haliza.katas@ukm.edu.my; 4Department of Chemical Sciences, Faculty of Science and Technology, Universiti Kebangsaan Malaysia, Selangor 43600, Malaysia; jalifah@ukm.edu.my; 5Centre for Drug and Herbal Development, Faculty of Pharmacy, Universiti Kebangsaan Malaysia, Kuala Lumpur 50300, Malaysia; juriyatijalil@ukm.edu.my

**Keywords:** cardiac, natural products, prevention, roselle, vessels, nutraceutical

## Abstract

Cardiovascular diseases (CVDs) represent a broad spectrum of diseases afflicting the heart and blood vessels and remain a major cause of death and disability worldwide. CVD progression is strongly associated with risk factors, including hypertension, hyperglycemia, dyslipidemia, oxidative stress, inflammation, fibrosis, and apoptosis. These risk factors lead to oxidative damage that results in various cardiovascular complications including endothelial dysfunctions, alterations in vascular integrity, the formation of atherosclerosis, as well as incorrigible cardiac remodeling. The use of conventional pharmacological therapy is one of the current preventive measures to control the development of CVDs. However, as undesirable side effects from drug use have become a recent issue, alternative treatment from natural products is being sought in medicinal plants and is gaining interest. Roselle (*Hibiscus sabdariffa* Linn.) has been reported to contain various bioactive compounds that exert anti-hyperlipidemia, anti-hyperglycemia, anti-hypertension, antioxidative, anti-inflammation, and anti-fibrosis effects. These properties of roselle, especially from its calyx, have relevance to its therapeutic and cardiovascular protection effects in humans. This review summarizes the findings of recent preclinical and clinical studies on roselle as a prophylactic and therapeutic agent in attenuating cardiovascular risk factors and associated mechanisms.

## 1. Introduction

Cardiovascular diseases (CVDs) present a significant burden to populations across the globe. Despite encouraging advances in prevention and treatments, CVDs remain a major cause of death and disability worldwide and relentlessly continue to grow over the years. More than 85 million people are living with CVDs in Europe [1], while more than half of all adults in the United States have some form of CVDs [2]. On the global scale, it was estimated that CVDs killed 17.9 million people in 2019, representing 32% of global deaths. Out of these deaths, 85% were due to heart attack and stroke [3]. Mitigating risk factors leading to the development of CVDs would be the best strategy for reducing CVD-related deaths. In fact, the World Health Organization (WHO) reported that reducing risk factors in young adults, in addition to maintaining an optimum risk profile through age 50, could prevent 90% of CVD events [4].

CVDs comprise a broad spectrum of diseases that can be categorized into different categories based on various criteria. Coronary heart disease, heart failure, myocardial infarction, cardiomyopathies, peripheral vascular diseases, hypertension, and peripheral vascular diseases are among the most reported CVDs [5]. The etiology of CVDs is complicated, and it involves several different abnormalities, such as metabolic abnormalities [6], genetic modifications [7], aberrant protein function [8], mitochondrial dysfunction [9], and uncontrolled cardiac remodeling [10]. Diabetes mellitus (DM), hypertension, and dyslipidemia are among the risk factors that contribute to the development of CVDs [5]. Moreover, these risk factors can trigger oxidative damage, inflammation, fibrosis, and apoptosis, which eventually aggravate the progression of CVDs. Hence, early and intensive control of CVD risk factors has the potential to markedly decrease cardiovascular events [11]. Therefore, there is an urgent need to mitigate the development of CVDs, and alleviating their risk factors would be the best shot at it.

Currently, conventional pharmacotherapy is a widely used and popular option to control the development of CVDs, but it can cause disturbances to the body due to its unfavorable effects. In this respect, ineffective treatment and management of CVD risk factors could eventually lead to myocardial infarction, heart failure, heart attack, stroke, and organ damage. Therefore, it is crucial to find an alternative treatment to manage the risk factors which eventually mitigate CVDs.

Natural products have shown to be an unrivaled source of novel and effective pharmaceuticals as they have enormous chemical diversity, therapeutic potential, safety, and cost-effectiveness, and are invaluable to be exploited as new therapeutic agents. Inevitably, recent findings have placed roselle (*Hibiscus sabdariffa* Linn.), a perennial plant of scarlet flowers native to South America and southern parts of Asia, as a prospective agent for CVD risk factor prevention and management [12,13]. Furthermore, roselle is a potent antioxidant as it is rich in bioactive compounds, including anthocyanins, caffeic acid, chlorogenic acid, ascorbic acid, and quercetin [14]. Roselle has been reported to possess anti-hyperlipidemic, anti-hyperglycemic, antioxidative, anti-inflammatory, anti-hypertensive, and anti-fibrosis effects, and therefore, roselle has the potential to be further explored as a pharmacotherapy agent in attenuating CVD risk factors. Thus, this review was performed to unveil the current knowledge of roselle and its roles in the prevention and treatment of CVD risk factors. The scientific literature search was conducted by employing the PubMed database as the search tool, which included appropriate keywords (roselle; hyperglycemia; hyperlipidemia; hypertension; oxidative stress; inflammation; fibrosis; apoptosis; cardiovascular; heart; vessel; *Hibiscus sabdariffa*). We summarized the articles and incorporated the literature search in accordance with the search results.

## 2. Origin and Beneficial Use of Roselle

Roselle, also known as red sorrel, is a tropical shrub of the Malvaceae family, particularly found in tropical regions such as India, Malaysia, Indonesia, Thailand, China, the Caribbean, Central America, West Africa, Asia, Brazil, Australia, Hawaii, and the Philippines [15]. Roselle has an acidic taste close to that of cranberry fruits. There are about 300 species of roselle that have been identified all over the world. This plant is commonly planted as an ornamental plant. However, due to the overwhelming utilization of roselle tree parts, including its calyx, leaves, stems, and seeds, in industries, roselle has now gained interest among researchers for its beneficial effects. Figure 1 shows a roselle plant.

Roselle has been traditionally used for decades as it has enormous benefits, mainly in medicinal and culinary aspects. The most common part of roselle that is widely used is the calyx. Fresh or dried calyces are used in the preparation of tea, either hot or cold, as well as fermented beverages, wines, jams, jellies, ice cream, aromatic agents, natural colorants, flavoring agents, and cakes [16]. Apart from that, roselle seeds are also commonly used to make oil, coffee substitutes, and are roasted and ground into a powder for seasoning in soups and sauces [17]. Roselle is a plant valued worldwide because all of its parts have medicinal uses thanks to various bioactive compounds that have significant effects, such as antioxidative, anti-inflammatory, anti-fibrosis, and anti-apoptosis effects, in attenuating various diseases. For decades, roselle has been traditionally used as folk medicine to treat urinary tract infections, toothaches, colds and fever, coughs, indigestion, and wounds, and to lower body temperature. Recent ethnomedicinal studies have shown that roselle consumption can be used as an anti-cancer, anti-microbial, anti-hyperlipidemic, anti-hyperglycemic, and anti-hypertensive treatment [14,18,19]. Roselle also has been widely used to detoxify aflatoxicosis [20]. Roselle has been proven to combat aflatoxicosis by attenuating toxicity-induced oxidative stress in the heart, which further limits CVD events.

### Bioactive Compounds of Roselle

Roselle is a plant that is rich in bioactive components, especially its calyx parts. Diverse phytochemical studies have validated that the major bioactive compounds found in roselle calyx are phenolic acids, flavonoids, anthocyanins, and organic acids, which are responsible for many biological activities [21,22,23]. Figure 2 demonstrates the bioactive compounds found in roselle.

Anthocyanins are the vital compounds that contribute to the bright color of roselle calyx and are rich in delphinidin-3-sambubioside, and cyanidin-3-sambubioside; meanwhile, delphinidin-3-glycoside and cyanidin-3-glycoside are present in smaller amounts [24,25]. Furthermore, several flavonoids have been identified in roselle calyx and leaves. Frequently reported flavonoids that are found abundantly are quercetin-3-glucoside, methyl epigallocatechin, myricetin, quercetin, rutin, and kaempferol [26]. Other types of flavonoids present in roselle include sabdaritrin, hibiscitrin, gossytrin, gossypitrin, luteolin, and pelargonidic acid [27,28]. In some studies, phenolic acid was noted to be present in roselle calyx, stems, and leaves. Protocatechuic and chlorogenic acids are essential phenolic acids and are present in higher amounts in roselle than other phenolic acids. Other phenolic acids in roselle are caffeic acid and gallic acid. Previous research has reported that total phenolic compounds were the most abundant in the calyx, followed by leaves and stems [29,30].

Moreover, roselle calyx has an acidic taste due to the high content of organic acid. Among the organic acids, hibiscus acid is the main compound that exists in roselle calyx, followed by citric acid, hydroxycitric acid, malic acid, and tartaric acid. Meanwhile, oxalic acid and ascorbic acid are found in smaller proportions compared to other organic acids [31,32]. Figure 2 demonstrates the bioactive compounds found in roselle. Roselle extract has also shown the presence of other secondary metabolites, including tannins, saponins, alkaloids, and glycosides that are responsible for contributing to biological activities. Several non-polar compounds also have been identified in roselle, particularly in seeds. The compounds include vitamin A (retinol), vitamin D, and vitamin E (α-tocopherol) [30].

The amount of each chemical constituent may vary depending on the condition used during the roselle extraction procedure. For instance, roselle polyphenol-rich extract contains hibiscus acid, chlorogenic acids, gallic acid, and quercetin, but no anthocyanin is detected in the extract [33]. This might be due to the high temperature used in the extraction process, which might cause anthocyanins to degrade. Additionally, total phenolic content was higher in the ethanol extract of roselle than in the aqueous extract [34]. When roselle calyx was extracted with citric acid, it was found that the total anthocyanin content was higher than when it was extracted with acidified ethanol. However, the same study found that the total phenolic content was higher when extracted with citric acid than acidified ethanol extract [35]. Nevertheless, the condition used for extraction should be considered to target the desired compounds from roselle extract.

## 3. Cardiovascular Risk Factors and Their Management

Over 70% of CVD cases and deaths in the world were attributed to risk factors [36]. Annual reports on regional and national CVDs and risk-related burdens may serve as the basis for developing effective strategies for CVD prevention [37]. Therefore, preventing and limiting the risk factors is an important step that should be undertaken to effectively reduce CVD-related mortality and morbidity [38]. Researchers have conducted extensive studies in order to find the best pharmacotherapy agents for CVD management by targeting risk factors, especially hypertension, diabetes mellitus, and dyslipidemia [39]. Moreover, focusing on the mechanisms that induce the occurrence of risk factors and progression of CVDs targeting oxidative stress, inflammation, fibrosis, and apoptosis is crucial to successfully control and limit CVD development [40,41,42].

CVDs are a chain of events initiated by related risk factors and progress through numerous pathophysiological pathways and processes to the development of end-stage cardiovascular disorder [43]. CVD risk factors, such as dyslipidemia, hypertension, and hyperglycemia, are known to promote oxidative stress and initiate a cascade of events, including inflammatory response, fibrosis, and apoptosis [44]. These mechanisms prompt endothelial damage, vascular stiffness, atherosclerosis, myocardial ischemia, and cardiomyocyte hypertrophy, which eventually lead to vascular and cardiac remodeling that culminates in CVDs [45,46].

Oxidative stress plays a crucial role in the progression of CVDs. Hyperglycemia results in the activation of alternative pathways of glucose metabolism, triggering the overproduction of reactive oxygen species (ROS) [47]. Moreover, hyperglycemia induces excessive fatty acid oxidation that leads to the derangement of free fatty acid generation [48]. Apart from that, hyperlipidemic conditions inaugurate beta-oxidation that promotes the increment in the electron transport chain and eventually accelerates mitochondrial dysfunction [49]. ROS generation can also be contributed by the activation of reduced nicotinamide adenine dinucleotide phosphate oxidase (NOX) derived from the stimulation of the renin–angiotensin–aldosterone system (RAAS) by hyperlipidemic and hyperglycemic conditions. The upregulation of NOX hampers the production of nitric oxide (NO) through the downregulation of endothelial nitric oxide synthase (eNOS) and further induces hypertension and promotes ROS generation [49]. Moreover, aflatoxicosis also can promote ROS production as aflatoxin can inhibit metabolic systems’ homeostasis and further effectuate oxidative stress [50,51].

The overproduction of ROS exceeds endogenous antioxidant capacity, leading to oxidative stress and the activation of inflammation, fibrosis, and apoptosis [52,53]. Oxidative stress further triggers inflammation by activating nuclear factor kappa B (NFκB), which upregulates proinflammatory genes and cytokine production and provokes the upregulation of transforming growth factor beta (TGF-β) and connective tissue growth factor (CTGF) that triggers the profibrotic response, which further causes extracellular matrix (ECM) accumulation, collagen deposition, and, in the end, fibrosis [54]. Oxidative stress, inflammation, and mitochondrial dysfunction trigger the cascade of apoptosis by enhancing the Bax/Bcl ratio, which prompts the activation of cytochrome c and caspase 3 that eventually lead to apoptosis [55]. These mechanisms ultimately cause cardiac hypertrophy, cardiac fibrosis, endothelial dysfunction, vascular stiffness, myocardial ischemia, and, ultimately, CVDs. Therefore, targeting oxidative stress, inflammation, fibrosis, and apoptosis in hypertension, dyslipidemia, and hyperglycemia that eventually contribute to CVDs remains a primary strategy in controlling and managing the development of CVDs. Figure 3 illustrates the association of risk factors and the mechanisms involved in the progression of CVDs.

Generally, the pharmacological management of CVDs involves controlling and managing the risk factors. Pharmacological therapy has shown explicitly that risk factor reduction decreases the risk of morbidity and mortality. For instance, treating blood pressure levels and lipid levels, particularly low-density lipoprotein (LDL) cholesterol, to the recommended levels has consistently been shown to reduce CVD events [56,57,58]. Managing normal glycemic levels by employing anti-hyperglycemic drugs helps reduce the risk of microvascular diseases such as retinopathy and nephropathy [59] in the short term while reducing risks for myocardial infarction and stroke [60,61]. In clinical trials, anti-hypertensive treatment, including diuretics, angiotensin-converting enzyme (ACE) inhibitors, angiotensin II receptor blockers, and calcium channel blockers, have effectively lowered blood pressure [62].

Despite a lot of conventional pharmacotherapy agents having been used for more than two decades, many studies have been conducted extensively in order to provide alternative treatment in the management of CVDs. Among the candidates, natural products have been proposed and widely investigated for their promising therapeutic effects, mainly on CVD risk factors. There exists broad consensus based on evidence from past studies that the consumption of natural products is rising around the world due to their proven effectiveness and because they can delay the progression of CVDs and are claimed to be safe to use [63].

## 4. Roselle in the Attenuation of Cardiovascular Diseases Risk Factors

Roselle is known as a plant that provides many health benefits, particularly as a source of antioxidants, and can serve as an alternative treatment for various diseases. Accumulated evidence has proven that roselle, mainly its calyx, contains many phytochemical compounds that are considered to be influential in providing various health benefits, especially as having anti-hyperlipidemic, anti-hyperglycemic, anti-hypertensive, antioxidative, anti-inflammatory, and anti-fibrosis effects. Hence, the utilization of roselle as an alternative treatment is crucial to alleviate CVD risk factors.

### 4.1. Anti-Hypertensive

Hypertension leads to CVD, which eventually causes death. A study reported that the consumption of 100, 200, and 400 mg/kg of aqueous extract of roselle leaves that contain saponins, tannins, flavonoids, alkaloids, phenols, and steroids for 6 weeks daily reduced systolic and diastolic blood pressure, mean arterial blood pressure, and heart rate in salt-induced hypertensive rats [64]. Furthermore, roselle aqueous extract that was mixed with several other plants, including chamomile, marjoram, and doum, successfully showed a significant reduction in blood pressure in N(gamma)-nitro-l-arginine methyl ester (L-NAME)-induced hypertensive animal models [65]. The same study suggested that the biological activities contributed by roselle were due to the presence of high concentrations of saponins, flavonoids, alkaloids, phenols, and tannins.

Nicotine is one of the culprits leading to hypertension. However, in one study, aqueous extracts of roselle treatments significantly decreased heart rate in a nicotine-induced heart damage model. The same study reported no difference in blood pressure. However, the treatment group showed a reduction trend in blood pressure [66]. In another study, the consumption of 100 mg/kg of roselle aqueous extract utilization rich in delphinidin-3-O-sambubioside and cyanidin-3-*O*-sambubioside for 28 days was revealed to effectively prevent the deterioration of cardiac systolic and diastolic function, coronary flow, as well as cardiac output in a diet-induced obesity rat model [13]. In a past study, roselle aqueous extract significantly ameliorated myocardial infarction-induced cardiac systolic and diastolic dysfunction, improved left ventricular developed pressure, improved isovolumetric relaxation, and attenuated angiotensin II-induced cardiomyocyte hypertrophy that eventually promoted hypotensive effects in animal models [67]. The same study indicated that these biological effects were due to the presence of ascorbic acid, chlorogenic acid, and caffeic acid in the roselle aqueous extract. Impaired vasorelaxation is a crucial event in the pathogenesis of hypertension.

In a diabetic cardiomyopathy model, polyphenol-rich extract of 100 mg/kg roselle supplementation for 8 weeks daily was shown to elevate left ventricular developed pressure and improve coronary flow, suggesting that roselle improves cardiac contractility and relaxation rate, which indicates that roselle has potential as a cardioprotective agent [68]. In another study, roselle polyphenol-rich extract containing a high concentration of flavonoids and phenolic acids was demonstrated to lower systolic function to the normal level, reduce heart rate, and improve coronary blood flow; all these positive effects were elicited by the pharmacological agonists for L-type Ca^2+^ channel, ryanodine receptor, B-adrenergic receptor, and sarcoendoplasmic reticulum calcium transport ATPase (SERCA) blocker, which were all abolished by roselle polyphenol extract in isolated rat heart [33].

One of the common pathways contributing to impaired vasorelaxation is an injury to the endothelium since it cannot produce relaxing factors such as nitric oxide. A previous study revealed that roselle aqueous extract supplementation with a dose of 100 mg/kg containing protocatechuic acid, hibiscus acid, delphinidin-3-sambubioside, cyanidin-3-sambubioside, and kaempferol-3-*O*-sambubioside for 28 days was found to restore vasorelaxation in aortic rings and exerted a vasodilatory effect by activating potassium channels, which causes hyperpolarization in vascular smooth muscle cells which then successfully lower systolic blood pressure in nicotine-induced vascular endothelial dysfunction [25]. Polyphenol-rich extract of roselle has been suggested to lower systolic and diastolic blood pressure mediated by the promotion of diuresis and ACE inhibitor activity of aorta in diabetic rat models given 100 mg/kg for 8 weeks, and it is believed to be mediated by delphinidin-3-O-sambubioside and cyanidin-3-*O*-sambubioside [69].

In an ex vivo study, treatment using methanolic extract of roselle calyx at a concentration of 10 ng/mL to 1 mg/mL in hypertensive rats was able to induce a vasodilator effect in isolated aortas through the activation of the nitric oxide/cGMP-relaxant pathway as well as the inhibition of Ca^2+^ influx, which explains the blood pressure-lowering effect by roselle [70]. Apart from that, the combination treatment of roselle with olive in hydro alcoholic powder with doses of 125, 250, and 500 mg/kg for 4 weeks resulted in a decrease in both systolic and diastolic blood pressure, reversed the L-NAME-induced suppression in serum nitric oxide, lowered the elevated ACE activity, enhanced eNOS gene and protein expression of the heart, and reduced aortic media thickness in a hypertensive rat model [71].

### 4.2. Anti-Hyperlipidemic

Several studies have shown that anti-hyperlipidemic treatment by roselle to reduce the level of cholesterol, especially targeting LDL cholesterol, can effectively prevent the risk of CVDs [72]. Aqueous extract of roselle supplementation of 100 mg/kg for 28 days significantly reduced the level of LDL cholesterol, plasma leptin, cholesterol, and triacylglycerol in diet-induced obesity rat models [13]. Moreover, alloxan-induced diabetic rats supplemented with methanolic extract of roselle seed in doses of 200 and 400 mg/kg for 2 weeks had considerably lowered concentrations of triglyceride, LDL cholesterol, and total cholesterol and increased high-density lipoprotein (HDL) cholesterol concentrations [73]. The same study revealed that the presence of tannins, saponins, glycosides, steroids, phlobatannins, flavonoids, phenols, and alkaloids contribute to the hypolipidemic effects of roselle methanolic extracts. Similarly, the administration of 100 mg/kg of polyphenol-rich extract of roselle for 4 weeks to type I diabetic rats significantly reduced plasma triglyceride, total cholesterol, and LDL cholesterol while increasing the level of HDL cholesterol [69,74].

A study by Farombi and Ige [75] revealed that the utilization of 100 and 200 mg/kg of ethanolic extract of roselle for 4 weeks demonstrated a better anti-hyperlipidemia effect compared to the dyslipidemia therapeutic drug, Lovastatin, in alloxan-induced diabetic rats. The same study demonstrated that roselle exhibited a decrease in cholesterol by 29%, very low-density lipoprotein (VLDL) cholesterol by 36%, LDL cholesterol by 40%, and atherogenic index by 32% compared to 5%, 23%, 28%, and 31% in rats treated with Lovastatin, respectively. Moreover, ethanolic extract of roselle leaves and calyxes containing anthocyanins, glycosides, protocatechuic acid, and hydroxycitric acid has been reported to lower total cholesterol, triglyceride, LDL, and VLDL cholesterol while increasing the level of serum HDL cholesterol in induced hyperlipidemic rats [76]. The same study hypothesized that the decrease in LDL cholesterol when consuming 500 mg/kg of ethanolic extract of roselle for 30 days is due to the capacity to inhibit the intestinal absorption of cholesterol, interfere with lipoprotein production, and induce hepatic LDL receptors’ expression, which could lead to the excessive removal of LDL cholesterol from the blood and increase the degradation and catabolism of cholesterol from the body.

Similarly, the consumption of polyphenol-rich extract of roselle significantly declined the serum level of LDL, triglyceride, and total cholesterol, as well as elevated HDL cholesterol levels in diabetic rat models [68]. On top of that, the utilization of roselle aqueous extract has been shown to reduce the atherogenic index, the coronary risk index, and LDL/HDL cholesterol significantly in type I diabetic rat models [12]. On the other hand, polyphenol extract and aqueous extract of roselle combined with a high-fat diet given to hamsters decreased cholesterol and triglyceride levels and suppressed the expression of fatty acid synthesis in hepatocytes, as well as lipid synthesis [77]. The same study indicates that the decrease in lipid content is due to the activation of 5’ adenosine monophosphate-activated protein kinase (AMPK) and the reduction in sterol regulatory element binding 1 (SREBP-1), thus inhibiting the expression of fatty acid synthase and 3-hydroxy-3-methylglutaryl coenzyme A (HMG-CoA) reductase.

### 4.3. Anti-Hyperglycemic

Roselle has significant anti-hyperglycemic effects. Aqueous extracts of red and white roselle were reported to reduce hyperglycemia through the inhibition of α-amylase and α-glucoside in an in vitro study as the enzymes responsible for carbohydrate hydrolysis, thus attenuating the occurrence of chronic diseases, mainly DM and CVDs [78]. Moreover, according to a study by Mardiah and colleagues [79], roselle aqueous extract given to type I diabetic rats at a dose of 288 mg/200 g body weight daily for 20 days reduced blood glucose due to the improvement of the number of β cells producing insulin in pancreatic cells. A study by Ajani et al. [12] proved that roselle aqueous extract was able to reduce the percentage of glucose in the blood in type I diabetic models given 200 and 500 mg/kg of the extract for 28 days.

Additionally, diabetic mice supplemented with 200, 400, and 600 mg/kg of ethanolic roselle calyx extract demonstrated a significant reduction in the level of blood glucose. The anti-hyperglycemic effect of roselle is attributed to bioflavonoids which mimic insulin by reducing blood glucose levels but also stimulate the secretion of insulin and the formation of glycogen in muscles [80]. Additionally, the administration of 0.1 and 1.0 g/kg daily of roselle ethanolic extract to streptozotocin-induced diabetic rats for 6 weeks significantly decreased fasting blood glucose levels, improved glucose tolerance, enhanced the basal release of insulin, and stimulated the regeneration of islets of Langerhans in the pancreas [81].

The methanolic seed extract of roselle decreased blood plasma glucose a week after diabetes induction, suggesting that roselle enhances the regeneration of β pancreatic cells [73]. Mashi et al. [82] reported that the antioxidant phenols and flavonoids might be responsible for the inhibition of blood glucose via insulin modulation, while El-Barky et al. [83] instead supported that saponin can induce the secretion of insulin from the pancreas or restore the insulin response or both to reduce plasma glucose. Furthermore, polyphenol-rich extract of roselle in diabetic rats significantly reduced plasma glucose [69].

### 4.4. Antioxidant

Roselle is well known for its antioxidant properties [29]. Morales-Luna et al. [84] reported that red and white roselle contains various phytochemical compounds, including organic acids (hibiscus acid), phenolic acids, anthocyanin, and flavonoids, with the capacity to inhibit the damaging effect of oxidative stress. For instance, aqueous extract of roselle fed to obese rats with myocardial infarction led to a significant reduction in oxidative stress by downregulating NOX2 and 8-isoprostane gene expression, and enhanced superoxide dismutase (SOD) enzyme activity and glutathione (GSH) concentration [13]. This is due to the ability of anthocyanins, phenolic acid, and flavonoids to inhibit oxidative stress by scavenging free radicals such as ROS and reactive nitrogen species (RNS) [84,85].

Furthermore, 100 kg/mL of aqueous roselle extract exhibited in vitro and in vivo the attenuation of cardiac oxidative stress by decreasing the gene expression of oxidative stress (NOX2 and 8-isoprostane level) and increasing the level of antioxidants (SOD and GSH level) in post-myocardial infarction rats [67]. Previously, Ramalingam and colleagues [86] revealed that supplementation with aqueous extract of roselle for 28 days prevented nicotine-induced myocardial oxidative stress by lowering the malonaldehyde (MDA) level, as well as by restoring activities of Cu/Zn-SOD, GSH content, and the GSH/GSSG ratio.

MDA, an oxidative marker of lipid peroxidation, correlates with the level of serum total cholesterol, triglyceride (TG), and LDL cholesterol [87], and extract leaves and calyces of roselle were proven to inhibit lipid peroxidation in vitro [76]. Moreover, atherosclerotic rabbit models that were fed delphinidin-3-*O*-glucoside from roselle in doses of 10 and 20 mg/kg for 12 weeks showed inhibited oxidative stress via the upregulation of GSH-PX and SOD1 mRNA levels [88].

Additionally, polyphenol-rich extract of roselle in diabetic rats significantly promoted the production of antioxidant markers SOD, GSH, and catalase (CAT) and attenuated the production of heart lipid peroxidation (TBARS), advanced oxidation protein product (AOPP), and MDA in the DM heart [69,74].

Mitochondrial damage is one of the key triggers that induce oxidative damage [9]. Previously, polyphenol-rich extract of roselle prevented mitochondrial-induced oxidative damage in DM hearts by decreasing the complex I activity of the mitochondrial respiratory chain [74]. Moreover, a decrease in mitochondrial structural damage and an increase in the mitochondrial antioxidant defense system in endothelial cells induced by DM was noted in diabetic rats fed a polyphenol-rich extract of roselle (Mohammed Yusof et al., 2018). The inhibition of lipid peroxidation and oxidative damage is associated with roselle’s Fe^2+^ chelating and OH^−^ radical scavenging properties and roselle’s reinforcement of the “mitochondrial” antioxidant system [74,78]. It can be concluded that roselle serves as a potent cardiovascular protective agent by attenuating ROS overexpression [69].

### 4.5. Anti-Inflammatory

Inflammation plays a critical role in the genesis, progression, and manifestation of CVD. Nevertheless, while safely modulating inflammation using targeted therapeutics remains a challenge, the results from past studies that utilize roselle as an alternative treatment demonstrate that targeting inflammation may offer the potential to reduce the risk for CVDs. A recent study conducted by Sun et al. [88] revealed that the supplementation of delphinidin-3-*O*-glucoside, a bioactive compound in roselle calyx, was able to reduce inflammation by downregulating the mRNA levels of interleukin (IL)-6, vascular cell adhesion molecule 1 (VCAM-1), and NF-κB in atherosclerotic rabbit models.

Moreover, the inflammation in a nicotine-induced cardiac injury animal model was successfully attenuated after being supplemented with aqueous roselle extract by preserving intracellular lactate dehydrogenase (LDH) activity in heart tissue [86]. Moreover, the utilization of 100 mg/kg roselle aqueous extract for 7 days has been proven to alleviate inflammation by attenuating the elevation of plasma troponin-T, IL-6, and IL-10, as well as downregulating IL-10 in rat models of myocardial infarction [89].

Previously, 500 mg/kg of a methanolic leaf extract from roselle fed to carrageenan-induced inflammation rats successfully decreased inflammation by decreasing inflamed paw diameter [90]. Similarly, a significantly higher inhibition of inflammation was also noted using roselle seed oil or extract compared to indomethacin on carrageenan-induced inflammation rats, indicating that the inhibition can be attributed to roselle’s ability to either inhibit the synthesis, release, or action of the histamine inflammatory mediators or antagonize the activity of the mediators after release and inhibit cyclooxygenase, lipoxygenase enzymes, and mast cell degranulation [91]. The same study stipulated that these biological effects were contributed by the presence of phenolic compounds, flavonoids, saponins, linalool, and alkaloids.

Furthermore, supplementation of petroleum ether extract of roselle seed greatly decreased the paw’s chronic and acute inflammation in a dose-dependent manner, inhibiting the inflammatory response of carrageenan in edema and the formation of granular tissue [92]. Roselle is also rich in the anti-inflammatory agent ascorbic acid and in antioxidant polyphenols that can reduce oxidation and promote the increase in the anti-inflammatory cytokine (IL10), therefore inhibiting proinflammatory cytokines (IL6 and tumor necrosis factor α (TNFα)) [93,94]. Hence, roselle possesses great antioxidant effects, which can inhibit inflammation in the cardiovascular and provide a cardioprotective effect.

### 4.6. Anti-Fibrosis

Fibrosis plays a fundamental role in the pathogenesis of heart failure and CVD. A body of growing evidence from the literature has demonstrated that roselle possesses protective effects against fibrosis. In the heart, extensive deposition of collagen and extracellular matrix end up stiffening and altering the physiological function of the heart. However, roselle aqueous extract treatment for 28 days ablated the deposition of myocardium collagen I and collagen III, and showed the ability to reduce the enlargement of cardiomyocytes in myocardial infarction rat models [67]. A previous study that was conducted by Si et al. [25] has shown that roselle treatments were able to reduce cardiac interstitial collagen accumulation, indicated by the reduction in B-type natriuretic peptide (BNP) gene expression, which is a cardiac fibrosis biomarker. Therefore, the reduction in cardiac fibrosis and BNP gene expression by roselle could improve ventricular compliance and relaxation in diet-induced obese animal models with myocardial infarction.

Furthermore, aqueous roselle extract has exerted anti-fibrotic activity by ameliorating the increase in collagen III gene expression as well as reduced collagen deposition in the heart with myocardial infarction [89]. Moreover, the combination treatment of roselle and olive with the doses of 125, 250, and 500 mg/kg for 4 weeks successfully attenuated extensive collagen fiber deposition and markedly reduced myocardial fibrosis in hypertensive animal models [71]. All of this accumulating evidence has displayed that roselle can be used as adjuvant therapy in ameliorating hypertension and later attenuating the progression of CVD. Table 1 summarizes the potential of roselle in the attenuation of CVD risk factors.

## 5. Clinical Studies

Roselle consumption has great potential to contribute to better human health. There is significant accumulating evidence in preclinical studies that corroborate roselle as a potent candidate for limiting CVD progression by targeting the risk factors and mechanisms involved [12,67]. In this regard, several clinical studies have been carried out to determine the effectiveness of roselle in enhancing human health, mainly by targeting CVD risk factors.

Some human trials have shown that roselle has exhibited positive effects in intervening cardiovascular risk factors. Recently, Harmili et al. [97] reported that the utilization of roselle as an intervention for hypertension patients for 7 days with a dose of 2 g every day has been shown to reduce systolic and diastolic blood pressure. Similar studies that were conducted on diabetic patients with mild hypertension have shown similar effects when consuming 240 mL of roselle infusion for a month and have revealed a decline in the systolic and diastolic blood pressure to the normal level [98]. In a quasi-experimental design study, the consumption of roselle flower tea twice a day for 2 weeks among hypertensive patients was able to reduce systolic and diastolic blood pressure [99].

A study has evaluated the effects of roselle flowers in the treatment of hypertension among postpartum mothers, and it was shown that consumption of 10 g brewed roselle with 200 mL of water together taken with anti-hypertensive drugs was able to reduce systolic and diastolic blood pressure when comparing before and after the intervention period [100]. Another previous study that was conducted in a double-blind, randomized controlled trial on sixty diabetic patients with mild hypertension who consumed roselle sour tea reported decreased systolic blood pressure and mean pulse pressure, highlighting the potential of roselle as an anti-hypertensive treatment [101].

Moreover, in a randomized, double-blind clinical trial, a standardized herbal medicinal product containing 1200 mg roselle that was consumed by hypertensive patients for 8 weeks led to a reduction in blood pressure, lowered TG, and reduced mean serum renin and ACE [102]. A non-randomized quasi-experimental study empirically corroborated that roselle has favorable cardiovascular effects in hypertensive patients, as 4 weeks of roselle ingestion with the dose of 250 mL every day was proven to bring down pulse pressure and heart rate to normal levels, and led to the regression of left ventricular hypertrophy [103]. Furthermore, 15 g of roselle infusion administered orally to people with hypertension for four weeks in a randomized clinical-case study was shown to have a lowering effect on arterial systolic and diastolic blood pressure and suppressed LDL levels, reduced cholesterol levels, enhanced HDL levels, and diminished TG levels [104]. This suggests that roselle is a potent candidate for hypertensive treatment.

Previously, a study that was conducted by Asgary et al. [105] in a double-blind placebo-controlled clinical trial showed that the daily consumption of 500 mg of roselle calyx powder for 4 weeks among forty adult patients with metabolic syndrome (hypertension, dyslipidemia, hyperglycemia) was able to significantly reduce serum TG and systolic blood pressure. Continuing with another randomized, double-blind, placebo-controlled trial, 300 mL of jelly drink containing polyphenol-rich roselle calyx extract given to forty-three adults with dyslipidemia for 8 weeks was demonstrated to decrease the levels of LDL-C, TG, TNFα, and MDA, and enhance the GSH level [87]. Hence, this study proved that roselle consumption could prevent CVD due to its polyphenolic content by exerting antioxidant and anti-inflammatory effects.

Additionally, Abubakar et al. [106] reported that a 250 mL polyphenol-rich roselle drink containing 7.5 g of roselle calyx powder, with 311 mg of gallic acid and 150 mg of anthocyanins, decreased postprandial systolic and diastolic blood pressure in subjects aged 47 to 49 with a cardiovascular risk of 1–10% after 4 h of consumption relative to baseline in a randomized, controlled, single-blinded, acute, and cross-over study. The same study also indicated that roselle consumption reduced serum glucose, plasma insulin, serum TG, and C-reactive protein (CRP) levels, and induced a significant improvement in the antioxidant response curve and no significant changes in arterial stiffness. The roselle extract improved postprandial vascular function and suggested a useful dietary strategy to reduce endothelial dysfunction and cardiovascular risk.

A study by Boushehri et al. [107] on the efficacy of sour roselle tea consumption in reducing selected CVD risk factors, involving about 362 participants in a systematic review and meta-analysis of randomized clinical trials, reported that sour roselle tea consumption significantly reduces fasting plasma glucose, systolic blood pressure, and diastolic blood pressure. Although drinking sour roselle tea had no significant influence on serum levels of triacylglycerol, total cholesterol, and HDL cholesterol, there was a tendency for a considerable reduction in LDL cholesterol serum concentrations. Another systematic review and meta-analysis on randomized controlled trials using roselle as an intervention for lipid profiles, blood pressure, and fasting plasma glucose levels in adult populations by Ellis et al. [108] indicated that roselle exerted stronger effects on an induced reduction in systolic and diastolic blood pressure and lowered LDL levels compared with other teas and a placebo.

Many clinical trials have been carried out to evaluate the effect of roselle in attenuating cardiovascular risk factors. No adverse effects were reported during the clinical trials. However, clinical studies exhibited some limitations, and the number of clinical studies is also still limited. Furthermore, treatment with roselle has addressed certain issues and highlighted why it is enigmatic to pursue clinical studies for large-scale clinical applications. For example, there is a lack of data related to the bioavailability of roselle phytochemicals during gastrointestinal digestion, which is crucial for maximizing the effects of roselle in alleviating CVD risk factors. Additionally, there is still limited research on the safety profile of roselle and the side effects, adverse effects, and toxicity associated with its consumption. Nowadays, the available roselle products on the market are not standardized. The isolation of bioactive compounds from roselle is time-consuming and costly, which impedes roselle-based product production on a large scale. Moreover, the stability of roselle-based products always faces obstacles regarding the stability of the physicochemical properties of the product, requiring a long process to access a proper evaluation of the stability to improve the product quality as well as increase both safety and efficacy. Thus, further studies are required in the future to validate the clinical efficacy of roselle in humans, mainly in alleviating cardiovascular risk factors, before proceeding with its large-scale clinical application. Table 2 summarizes the clinical studies that use roselle as treatment. Figure 4 proposes a summary of the role of roselle in attenuating CVD by targeting its risk factors and the mechanisms involved.

## 6. Conclusions

The risk factors of CVD, especially hyperglycemia, hypertension, and hyperlipidemia, are the main culprits that induce the development of CVD via oxidative stress, inflammation, fibrosis, and apoptosis mechanisms. Roselle has been studied for decades due to its promising effects in limiting the CVD progression by exhibiting anti-hyperglycemic, anti-hyperlipidemic, anti-hypertensive, antioxidative, anti-inflammatory, anti-fibrosis, and anti-apoptosis effects in preclinical studies and clinical studies. The main metabolites in roselle that play a significant role in these biological effects are anthocyanins, phenolic acids, flavonoids, organic acids, saponins, and tannins. However, most of the studies that were conducted do not highlight which compound found from each type of extract shows potent results in alleviating CVD risk factors. Hence, future studies should consider this matter so that the biological properties of roselle can be fully utilized. Furthermore, limited studies have been conducted to develop roselle as a nutraceutical product. Therefore, this review can provide knowledge about roselle for developing nutraceuticals in the future, mainly targeting CVDs.

## Figures and Tables

**Figure 1 pharmaceuticals-16-00807-f001:**
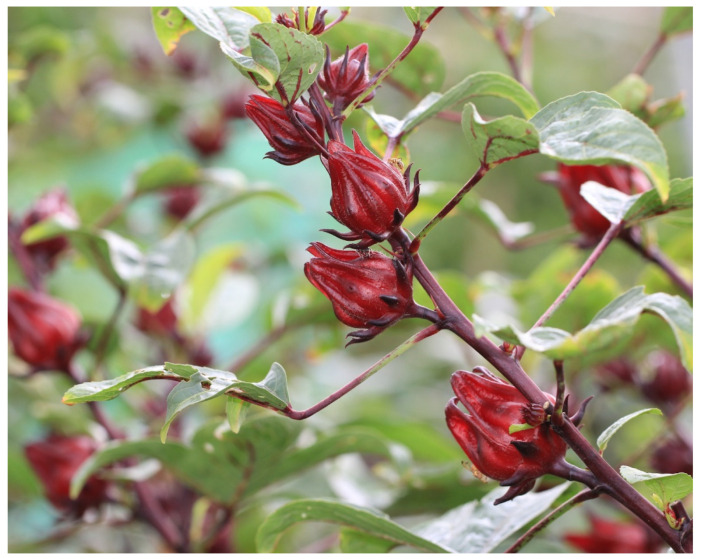
Roselle plant.

**Figure 2 pharmaceuticals-16-00807-f002:**
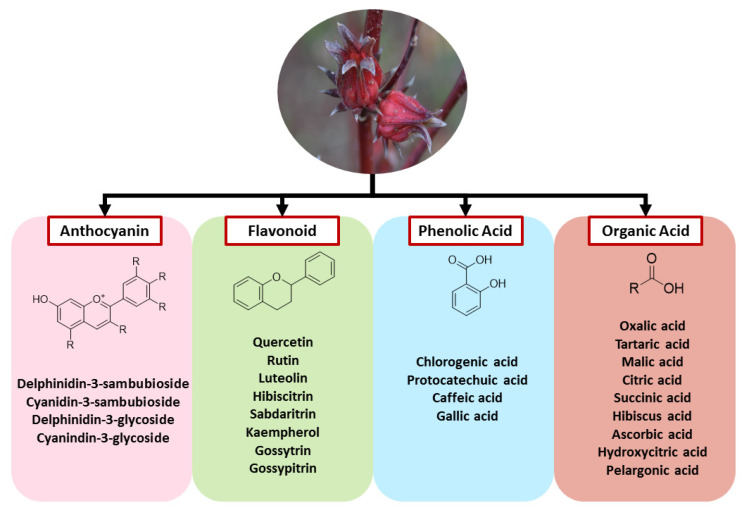
The bioactive compounds found in roselle.

**Figure 3 pharmaceuticals-16-00807-f003:**
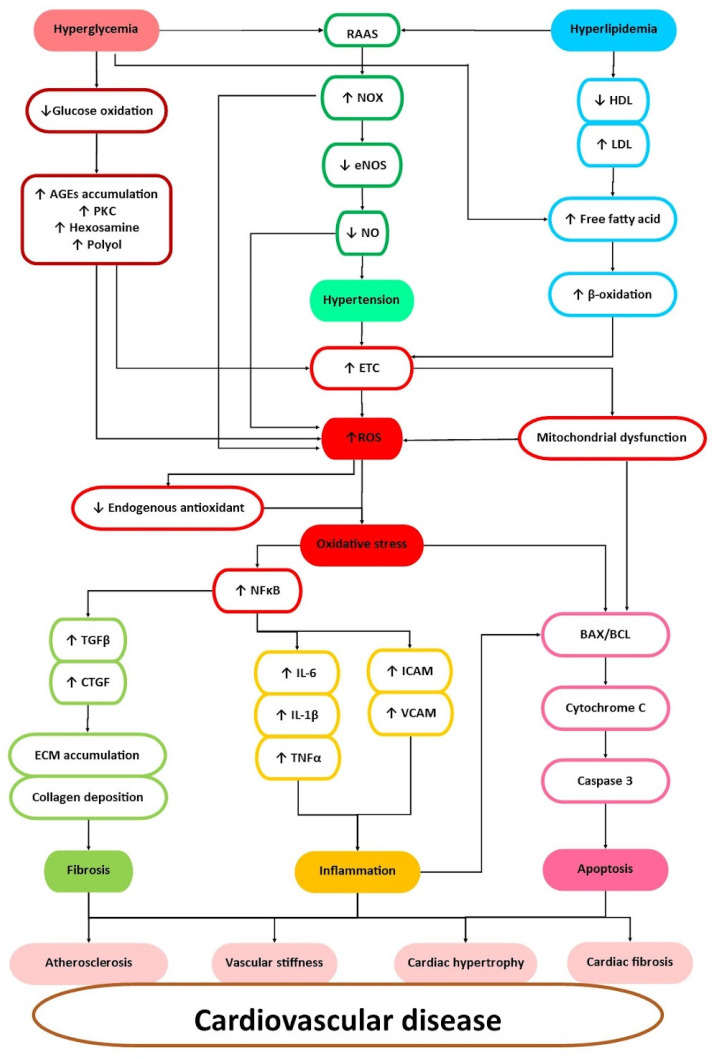
The association of risk factors and the mechanisms involved in the progression of CVD. Hyperglycemia reduces glucose oxidation and activates alternative pathways of AGE accumulation, polyol, hexosamine, and PKC pathways that contribute to ROS. Hyperlipidemia also contributes to the elevation of ROS production via lower HDL, enhances LDL that eventually promotes an increase in free fatty acid accumulation and activates beta-oxidation, and induces an increment in electron transfer chain and mitochondrial dysfunction. Both hyperglycemia and hyperlipidemia promote RAAS activation and lead to the upregulation of NOX and then the downregulation of eNOS, which lead to a reduced NO level. This will cause vasoconstriction and hypertension. The upregulation of NOX further contributes to ROS generation. The overproduction of ROS hampers endogenous antioxidants and causes oxidative stress. Oxidative damage leads to the activation of NFkB and promotes the elevation of proinflammatory cytokines (IL-6, IL-1β, TNF-α, ICAM, VCAM) and profibrotic factors (TGF-β, CTGF) that lead to inflammation and fibrosis (ECM accumulation, collagen deposition). Oxidative stress and inflammation further trigger apoptosis via Bax/Bcl, cytochrome c, and caspase 3. All of the mechanisms eventually lead to endothelial dysfunction, vascular stiffness, cardiac hypertrophy, cardiac fibrosis, and, later, CVD. (Abbreviations: AGE—Advanced Glycation End Product; PKC—Protein Kinase C; RAAS—Renin Angiotensin Aldosterone System; NOX—NADPH Oxidase; eNOS—Endothelial Nitric Oxide Synthase; NO—Nitric Oxide; ROS—Reactive Oxygen Species; ETC—Electron Transport Chain (ETC); NFκB—Nuclear Factor Kappa Light Chain Enhancer of Activated B cells; HDL—High-Density Lipoprotein; LDL—Low-Density Lipoprotein; TGF-β—Transforming Growth Factor-β; CTGF—Connective Tissue Growth Factor; IL—Interleukin; TNF-α—Tumor Necrosis Factor Alpha; ICAM—Intercellular Adhesion Molecule; VCAM—Vascular Cell Adhesion Molecule 1.) ↑ indicates upregulation/increase meanwhile ↓ indicates downregulation/decrease.

**Figure 4 pharmaceuticals-16-00807-f004:**
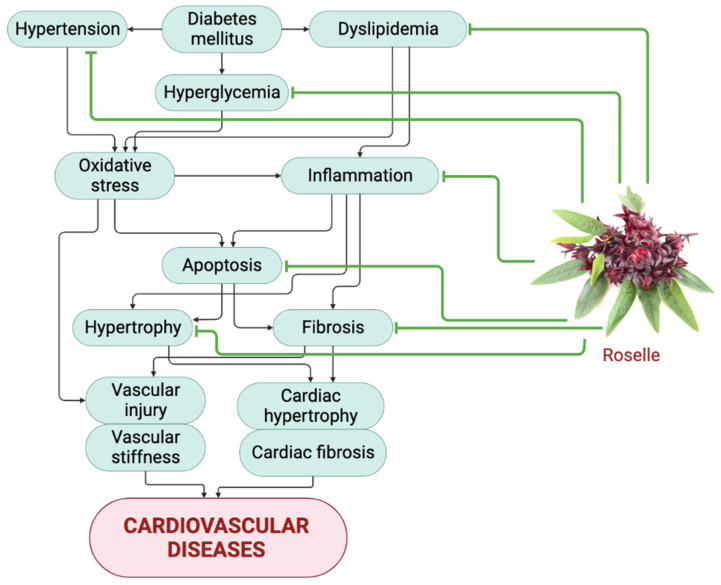
Summary of the role of roselle in limiting the development of CVD. Roselle was able to alleviate the progression of CVD by targeting its risk factors, especially hyperglycemia, dyslipidemia, and hypertension, as well as the involved mechanisms, including oxidative stress, inflammation, fibrosis, and apoptosis. The green line indicates the activity of roselle meanwhile the black color line indicates the mechanism that lead to cardiovascular diseases.

**Table 1 pharmaceuticals-16-00807-t001:** Summary of roselle usage in the attenuation of cardiovascular disease.

Activity	Roselle Part	Extract Type	Dose	Duration of the Study	Study Model	Mechanism of Action	References
Anti-hypertensive	Leaves	Aqueous extract	100, 200, 400 mg/kg	6 weeks	In vivo	Reduced systolic and diastolic blood pressure, mean arterial blood pressure, and heart rate.	[64]
	Calyx	Aqueous extract	13.347, 26.694 mg/kg	4 weeks	In vivo	Prevented an increase in blood pressure.	[65]
	Calyces	Aqueous extract	100 mg/kg	21 days	In vivo	Decreased blood pressure and heart rate in nicotine-exposed rats.	[66]
	Calyx	Aqueous extract	100 mg/kg	28 days	In vivo	Exerted vasodilatory effects on the aortic ring, activated potassium channel, and reduced blood pressure.	[25]
	Calyx	Hydro-alcoholic powdered extract	125, 250, 500 mg/kg (4 weeks)	4 weeks	In vivo	Decreased systolic and diastolic blood pressure, reversed the suppression of nitric oxide, lowered ACE activity, and enhanced eNOS gene and protein expression.	[71]
	Calyx	Methanol extract	10 ng/mL to 1 mg/mL	-	Ex vivo	Induced vasodilator effect in isolated aortas via nitric oxide/cGMP-relaxant pathway and inhibited Ca^2+^ influx.	[70]
	Calyx	Aqueous extract	100 mg/kg	28 days	In vivo	Ameliorated cardiac systolic and diastolic dysfunction; improved coronary flow and cardiac output.	[13]
	Calyx	Roselle polyphenol extract	125 to 2000 µg/mL	-	Ex vivo	Lowered systolic function, reduced heart rate, increased relaxation, and improved coronary blood flow.	[33]
	Calyx	Aqueous extract	100 mg/kg	28 days	In vivo and in vitro	Ameliorated myocardial infarction-induced cardiac systolic and diastolic dysfunction, improved left ventricular developed pressure, attenuated angiotensin II-induced cardiomyocyte hypertrophy.	[67]
	Calyx	Polyphenol-rich Extract	100 mg/kg	8 weeks	In vivo	Exerted hypotensive effects by reducing systolic blood pressure.	[69]
	Calyx	Polyphenol-rich extract	100 mg/kg	8 weeks	In vivo	Improved cardiac contractility and relaxation rate by improving left ventricular developed pressure and coronary blood flow.	[68]
Anti-hyperlipidemic	Calyces	Aqueous	100 mg/kg	28 days	In vivo	Reduced the level of LDL-C in the blood through LDL hydrolysis and lipolysis.	[13]
	Calyces and leaves	Ethanolic extract	500 mg/kg	30 days	In vivo	Inhibited the intestinal absorption of cholesterol, interfered with lipoprotein production, and induced hepatic LDL receptor expression.	[76]
	Flower	Ethanolic	100 and 200 mg/kg	28 days	In vivo	Reduced cholesterol by 29%, VLDL-C by 36%, LDL-C by 40%, and atherogenic index by 32%.	[75]
	Calyces	Polyphenol-rich extract	100 mg/kg	8 weeks	In vivo	Reduced serum levels of LDL/HDL and TG/HDL	[68]
	Seed	Methanol extract	400, 200 mg/kg	14 days	In vivo	Lowered the concentration of TG, LDL-C, and cholesterol while increasing the HDL concentration.	[73]
	Calyces	Polyphenol-rich extract	100 mg/kg	28 days	In vivo	Reduced serum level of LDL-C and increased the level of HDL-C.	[74]
	Calyces	Polyphenol-rich extract (HPE)	100 mg/kg	8 weeks	In vivo	Reduced plasma TG, cholesterol, and LDL-C, and increased level of HDL-C.	[69]
	Leaves	Lyophilized extract and polyphenols	1%, 2%, 0.1%, 0.2% (10 weeks)	10 weeks	In vivo and in vitro	Suppressed the expression of fatty acid synthesis in hepatocytes; inhibited hepatic intracellular lipid accumulation and lipid synthesis.	[77]
Anti-hyperglycemic	Calyces (white and red)	Aqueous	0–200 µL	-	In vitro	Reduced hyperglycemia through inhibition of α-amylase and α-glucoside; inhibited the breakdown of starch into oligosaccharides and monosaccharides.	[78]
	Calyces	Ethanol	0–37.5 µL	-	In vitro	Reduced the concentration of α-amylase and α and β-glucosidase in a concentration-dependent manner, inhibited glucosidase, and disrupted hydrolysis and the release of glucose.	[95]
	Calyces	N-hexane, ethyl acetate, and ethanol extract	200, 400, and 600 mg/kg	-	In vivo	Bioflavonoids in roselle mimic insulin by inhibiting blood glucose levels and stimulating insulin secretion and formation of glycogen in muscle cells.	[80]
	Calyces	Polyphenol-rich extract	100 mg/kg	28 days	In vivo	Reduced plasma glucose and dyslipidemia.	[74]
	Seed	Methanol extract	400, 200 mg/kg	14 days	In vivo	Decreased plasma glucose through regeneration of β-pancreatic cells. Saponins in the roselle lower blood glucose through the restoration of insulin response or induction of insulin release from the pancreas.	[73]
Antioxidative	Calyces	Aqueous and methanolic	750 and 500 mg/100 mL potable water ad libitum	16 weeks	In vivo	Organic acids (hibiscus acid), phenolic acids, anthocyanin, and flavonoids inhibit the damaging effect of oxidative stress.	[84]
	Calyces	Aqueous	100 mg/kg	28 days	In vivo	Reduced *NOX2* gene expression and 8-isoprostane level; increased the level of antioxidants SOD and GSH.	[13]
	Calyces	Polyphenol-rich extract	100 mg/kg	8 weeks	In vivo	Inhibited MDA and AOPP in the aorta with notably increased levels of GSH antioxidants.	[96]
	Calyces	Polyphenol-rich extract	100 mg/kg	8 weeks	In vivo	Reduced MDA and AOPP levels. Enhanced the production of GSH.	[69]
	Calyces	Polyphenol-rich extract	100 mg/kg	4 weeks	In vivo	Attenuated oxidative stress and cellular damage; decreased the mitochondrial complex I activity.	[74]
	Calyces	Polyphenol-rich extract	100 mg/kg (28 days)	8 weeks	In vivo	Decreased the mitochondrial structural damage and increased the overexpression of MnSOD or SOD2 in mitochondria.	[68]
	White and red calyces	Aqueous	0–200 µL	-	In vivo	Roselle chelates Fe^2+^ and scavenges OH^−^ radical. Decreased lipid peroxidation and oxidative damage in the pancreas.	[78]
	Calyces	Polyphenol-rich extract	100 mg/kg	28 days	In vivo	Enhanced the production of SOD, SOD2, GSH, and CAT; attenuated the production of heart lipid peroxidation (TBARS) and (AOPP).	[74]
	Calyces	Aqueous	100 mg/kg	28 days	In vivo and in vitro	Inhibited the gene expression of oxidative stress (NOX2 and 8-isoprostane level) and increased the level of antioxidant SOD and GSH in post-myocardial infarction rats.	[67]
	Isolated delphinidin-3-O-sambubioside	-	10 and 20 mg/kg	12 weeks	In vivo	Upregulated the mRNA expression of aortic GSH-PX and SOD1.	[88]
	Calyces	Aqueous	100 mg/kg	28 days	In vivo	Enhanced the production of Cu/Zn-SOD and GSH as well as the GSH/GSSC ratio.	[86]
Anti-inflammatory	Isolated delphinidin-3-O-sambubioside	-	10 and 20 mg/kg	12 weeks	In vivo	Inhibited the production of aortic ICAM-1, MCP-1, VCAM-1, CRP, IL-6, and TNF-α.	[88]
	Calyces	Aqueous	100 mg/kg	28 days	In vivo	Enhanced the intracellular LDH activity in the heart after nicotine administration.	[86]
	Calyces	Aqueous	100 mg/kg	10 days	In vivo	Reduced the plasma level of troponin-T, IL-6, and IL-10 and attenuated the expression of IL-10 in MI rats.	[89]
	Leaves	Methanolic	250 and 500 mg/kg		In vivo	Decreased paw diameter.	[90]
	Seed	Oil and extract	500 mg/kg	5 h	In vivo	Inhibited inflammation by reducing the synthesis, release, or action of the inflammatory mediators or antagonizing the activity of the mediators after release.	[91]
	Seed	Petroleum ether	2–8 mL/kg		In vivo	Reduced chronic and acute inflammation in a dose-dependent manner through suppression of cyclooxygenase and prostaglandin synthesis.	[92]
	Flower petals	Ethanol	300 mg/kg	28 days	In vivo	Promoted the increase in IL10 production and inhibited the synthesis of IL6 and TNF-α production.	[93]
Anti-fibrosis	Calyx	Aqueous extract	100 mg/kg	28 days	In vivo and in vitro	Suppressed the production of collagen I and collagen III in myocardial infarction.	[67]
	Calyx	Aqueous extract	100 mg/kg	28 days	In vivo	Decreased interstitial collagen deposition and downregulated BNP.	[13]
	Calyx	Aqueous extract	100 mg/kg	10 days	In vivo	Reduced collagen deposition; downregulated collagen III expression.	[67]
	Calyx	Hydro-alcoholic powdered extract	125, 250, 500 mg/kg	4 weeks	In vivo	Attenuated extensive collagen fiber deposition and myocardial fibrosis.	[71]

**Table 2 pharmaceuticals-16-00807-t002:** Summary of clinical studies using roselle as treatments.

Roselle Part	Study Design	Subjects	Dose	Treatment Duration	Effects	References
Calyx powder	Double-blind, placebo-controlled	40 adult patients with metabolic syndrome	500 mg/day	4 weeks	Reduced serum TG and systolic blood pressure	[105]
Polyphenol-rich roselle calyx extract in the jelly drink	Randomized, double-blind, placebo-controlled trial	42 adults with dyslipidemia	300 mL/day	8 weeks	Decreased levels of LDL-C, TNFα, and MDA and increased GSH level	[87]
Standardized herbal medicinal product containing roselle	Phase II, randomized, double-blind, captopril-controlled clinical trial	134 patients with grade 1 essential hypertension	1200 mg	8 weeks	Reduced blood pressure, lowered TG, reduced mean serum renin and ACE	[102]
Roselle sour tea	Double-blind, randomized controlled trial	60 diabetic patients with mild hypertension	2 g of roselle tea in 240 mL boiling water two times/day	1 month	Decreased systolic blood pressure and mean pulse pressure	[101]
Roselle infusion	Non-randomized, quasi-experimental study	50 subjects with moderate essential hypertension	2 g in 250 mL of boiling water/day	4 weeks	Lowered pulse pressure and heart rate, reduced ventricular hypertrophy	[103]
Roselle infusion	Prospective randomized clinical-case study	24 hypertensive patients	15 g/day	4 weeks	Lowered arterial systolic and diastolic blood pressure, suppressed LDL level, reduced cholesterol level, enhanced HDL level, diminished TG level	[104]
Roselle calyx infusion	Quasi-experimental with a pre-test and post-test nonequivalent control group	17 people with hypertension	2 g roselle-infused drink/day	7 days	Reduced systolic and diastolic blood pressure	[97]
Roselle calyx infusion	Randomized clinical case study	40 diabetic patients with mild hypertension	240 mL of roselle two times/day	1 month	Reduced systolic and diastolic blood pressure	[98]
Roselle flower tea	A quasi-experimental design with a two-group pre-test and post-test design	16 patients with hypertension		2 weeks		[99]
Roselle dried flower infusion	A quasi-experimental study with a nonequivalent control group design	30 postpartum mothers with hypertension	10 g brewed with 200 mL water/day	6 h	Reduced systolic and diastolic blood pressure	[100]
Roselle calyx tea	Randomized, controlled, single-blinded, 2-meal cross-over study	25 participants with 1 to 10% CVD risk factors	7.5 g calyx in 250 mL Buxton water	4 h	Decreased postprandial systolic and diastolic blood pressure, reduced serum glucose, plasma insulin, serum TG, CRP level, improved antioxidant response curve	[106]
Roselle sour tea	Meta-analysis, randomized clinical trials	362 participants with hypertension	-	-	Reduced fasting plasma glucose, lowered systolic and diastolic blood pressure	[107]
Roselle tea	Meta-analysis, meta-regression	415 participants with hypertension and cardiometabolic markers	-	-	Reduced systolic and diastolic blood pressure, lowered LDL level	[108]

## Data Availability

Data sharing not applicable.

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
