# Peer review of "Therapeutic Potential of Hibiscus sabdariffa Linn. in Attenuating Cardiovascular Risk Factors"

_pharmaceuticals, 2023, doi:10.3390/ph16060807_

Round 1
Reviewer 1 Report
The problem of cardiovascular diseases is of great importance in most countries in the world, especially in developed countries. Therefore, research on the possibility of CVD prevention and treatment with the use of various pharmaceuticals, including those of natural origin, is extremely important. For these reasons, the presented review is very interesting and worth attention. Figures/schemes and tables enrich the article. The review seems to be complete and provides a broad overview of the subject. Below are my minor comments for the Authors.
1. Latin names throughout the manuscript should be in italics.
2. All figures should be cited in the text with a comment/description (see line 90, Figure 1. is not enough).
3. Lines 115-130: English in this paragraph should be corrected.
4. Line 354: it should be Antioxidant.
5. In my opinion, the list of abbreviations could be an additional help for readers, but every abbreviation should be explained directly in the text while first used.
6. Conclusion - English of this paragraph should be corrected.
The English language in the article is mostly very good, only some parts seem to be written not entirely correctly. It is worth reviewing the manuscript again to standardize the quality of the English language throughout the manuscript.
Reviewer 2 Report
Dear authors
I carefully checked your review and the content was interesting. However, please address the following comments within the manuscript and revise the content based on the given notes.
1- Please add search strategies to collect scientific literature used in preparing this manuscript. Please add this section at the end of the introduction.
2- Please italicize scientific names.
3- Please make sure all represented figures are original. If copyright materials are used in preparing this study, please add a license agreement for each file.
4- Please check that problematic papers such as retracted papers were not cited within the manuscript reference list.
5- Please add the negative effects of the studied plant to the study if any.
6- Please amend some misspelled words within the text.
7- Please don’t repeat the name of the plant genus after citing its scientific name for the first time.
8- Which metabolite plays a significant role in the biological activity of this plant?
9- It has been proved that the consumption of aflatoxin-contaminated foods is associated with the risk of cardiovascular diseases and cancer. Please add some paragraphs about the cross-links between the consumption of Roselle and the prevention of the health complication of aflatoxins. Due to the presence of polyphenols in this plant, it seems clear that this plant has the potential to be considered as a supplementary agent in preventing aflatoxin-mediated health complications. You can check the following manuscripts and discuss this case regarding the biological activity of Roselle against aflatoxin contaminations.
https://www.sciencedirect.com/science/article/abs/pii/S095671351100404X
https://www.frontiersin.org/articles/10.3389/fnut.2022.981984/full
10- Please add a paragraph regarding the clinical limitation of Roselle for large-scale applications.
The rest of the manuscript seems interesting and I have no further comments on this paper. Please carefully revise your paper and amend the content of your perfect manuscript based on the given points.
The English language is good and it requires a minor revision to amend some misspelled words within the paper.
Reviewer 3 Report
The reviewed paper is focused on Hibiscus sabdariffa, as a prospective agent for cardiovascular diseases (CVDs) risk factors prevention and management. The authors showed information on CVDs including cardiovascular risk factors, bioactive compounds present in roselle, and review of recent studies of preclinical and clinical studies on this plant. The paper is quite well written, but in my opinion it requires some modification in order to be considered for publication in Molecules. The modification are necessary in case of bioactive compounds.
Section 2.1.
In this section we are able to read about the bioactive compounds present in Hibiscus sabdariffa. But the authors were focus on only the major components, and only calyx. We can read that “the major bioactive compounds that are found in roselle calyx are phenolic acids, flavonoids, anthocyanins, and organic acids which are responsible for many biological activities” The mentioned four group of compounds are somehow described. However, in the following sections we can read about the active extracts obtained not only from calyx, but also leaves, flowers, seeds. We can also read about the active petroleum ether extract, which is non-polar, but the mentioned group of compounds are rather polar one. SO, some additional description is necessary.
Lines 120-122: We can read that other types of flavonoid present in roselle including eugenol and phytosterol. Both are not flawonoids!!!
Section 4.1: We can read that “…aqueous extract of roselle’s leaves that contain saponins, tannins, flavonoids, alkaloids, phenols, and steroids”. In this fragment not previously mentioned roselle metabolites appeared. These are saponins, alkaloids and tannins. Please provide more detail about these metabolites, but in section 2.1.
Section 4.2: one more group of compounds appeared in this section. These are phlorotannins. Are the authors sure that phlorotannins can be found in hibiscus? As far as I know these metabolites are characteristic for algae and seaweeds.
